# Leveraging Deep Learning Techniques and Integrated Omics Data for Tailored Treatment of Breast Cancer

**DOI:** 10.3390/jpm12050674

**Published:** 2022-04-22

**Authors:** Deeba Khan, Seema Shedole

**Affiliations:** Department of Computer Science and Engineering, Ramaiah Institute of Technology, Bengaluru 560054, India; seemas@msrit.edu

**Keywords:** breast cancer subtype, deep learning, multi-omics data, drug response, basal-like breast cancer

## Abstract

Multiomics data of cancer patients and cell lines, in synergy with deep learning techniques, have aided in unravelling predictive problems related to cancer research and treatment. However, there is still room for improvement in the performance of the existing models based on the aforementioned combination. In this work, we propose two models that complement the treatment of breast cancer patients. First, we discuss our deep learning-based model for breast cancer subtype classification. Second, we propose DCNN-DR, a deep convolute.ion neural network-drug response method for predicting the effectiveness of drugs on in vitro and in vivo breast cancer datasets. Finally, we applied DCNN-DR for predicting effective drugs for the basal-like breast cancer subtype and validated the results with the information available in the literature. The models proposed use late integration methods and have fairly better predictive performance compared to the existing methods. We use the Pearson correlation coefficient and accuracy as the performance measures for the regression and classification models, respectively.

## 1. Introduction

Of the many types of cancer, breast cancer (BC) is the most prevalent cancer among women across the globe. It is increasing at the alarming rate of 14% each year in India. Fifty-seven percent of the total cancers reported were BC, making it the leading [1,2] type of cancer. Globocan’s 2020 [3] data suggest that “female breast cancer is the second most commonly diagnosed cancer, with an estimated 2.3 million new cases (11.7%) and a death rate of 6%”. These global data also reveal that one in 28 women is likely to be affected by BC during her lifetime. This statistic suggests a need to conduct research related to the diagnosis and treatment of BC. Although terminal diseases such as cancer have been the leading cause of death, intensive research leading to timely interventions and personalized therapies has improved the prognosis [4,5]. The main factors that influence personalized treatment are the genotype of the patient and the patient’s history and sensitivity to the therapy administered. Only 15% of genomic-based studies have regulatory consent [6]. Although many knowledge-based systems have aided in improved therapies, there is still room for improvement in these intelligent systems for better prognosis of BC patients. The systems can leverage the patterns from varied omics along with phenotypic information to address the gaps in the effective application of personalized treatment.

To profile heterogeneous genotype data related to BC, high-throughput technologies may be exploited. Genotype data, precisely known as genomics data comprising the gene profiles of patients or cell lines, are hosted by popular websites such as The Cancer Genome Atlas (TCGA) and the Cancer Cell Line Encyclopedia (CCLE) [7,8]. The genomic data at their disposal are in the form of miRNA gene expression, mRNA gene expression, somatic mutation, copy number variation, methylation, etc. Early studies were based on the analysis of single omics data, such as mRNA-based gene expression or DNA methylation. This single-omics analysis is limited to exploring the underlying biological mechanisms [9] and capturing the intricacy for various complex diseases that can explain its molecular property. Recent studies have shown that integrating omics data gives a better understanding of the overall impact that can influence cancer profiling, diagnosis, and treatment. This integration has helped in identifying biomarkers and predicting clinical outcomes for various genetic diseases, such as cancer and Alzheimer’s disease. Modelling integrated omics data for predicting cancer subtypes and predicting the drug responses of any cancer can be a game changer for personalized medicine. Highly available multiomics data along with high-performance computational methods and machine learning algorithms can be effectively leveraged in addressing challenges related to personalized medicine.

In this paper, we propose methods to model multiomics data for classifying BC into subtypes and DCNN-DR for predicting the drug response of cancer drugs on BC in particular. Deep learning techniques such as DNNs, CNNs and autoencoders are exploited. The performance of the models is compared with the existing methods for similar purposes. Furthermore, we also used DCNN-DR to predict effective drugs for BLBC, a subtype of BC.

### 1.1. BC Subtypes

The classification of BC patients into proper subtypes is important for selecting an appropriate line of treatment. Moreover, having prior knowledge of how each subtype is distinguished is vital for research into new treatments. The molecular classification of BC was introduced in 1999 by the National Cancer Institute (NCI). Initially, Perou et al. [10] proposed four BC subtypes: the luminal subtype, the basal-like subtype, the human epidermal growth subtype and the normal breast-like subtype. The luminal subtype was further classified by Sorlie et al. [11] as luminal A and luminal B. Waks et al. [12] classified BC into three major subtypes based on hormone bimolecular markers, oestrogen receptor (ER), progesterone receptor (PR) and HER2 (hormone receptor 2): ER+/PR+/HER2−, HER2+ and TNBC. Tao et al. [13] proposed five subtypes based on histological markers along with ER, PR and HER2. These subtypes are luminal A, luminal B, HER2+ and unclear type. In our work, we consider five subtypes based on PAM and TCGA [14,15]. The summarized information of the five subtypes considered is found in Table 1.

Luminal A accounts for 60% of all BC types and has the highest survival rate [16]. Elderly patients are at risk of luminal B. The HER2+ subtype accounts for 25% of total BC with a poor survival rate. Classification is essential to decide the line of treatment and therapies used. For example, luminal A and B patients are sensitive to endocrine therapy. Patients with Her2-enriched and normal-like subtypes benefit from Her2 and Ki-67 targeted therapy, respectively. Basal-like subtypes pose challenges to clinicians as there is no targeted therapy yet, due to its heterogeneity [17]. The treatments are usually in conjunction with radiation, chemotherapy and surgery, depending on each case. Early subtype classifications were based on immune-histological analysis, mammography and radiology. With the high availability of omics data, models have been proposed for classification based on molecular data. This approach can complement immunohistochemistry techniques for better diagnosis. The earlier models are based on single omics data analysis, such as gene expression or DNA methylation. Both supervised and unsupervised methods were proposed. Recently, models have been developed by integrating omics data and using machine learning techniques such as SVM and deep learning [13,18]. In our study, we used omics data to represent different characteristics of gene profiles, integrate them and subsequently analyze them. The subset of features from each omics dataset is combined to form the input for the autoencoder that forms the basis of classification.

The multiomics data used in our study are copy number variation, mutation, methylation, miRNA, RNA, and protein-expression [19]. Clinical data were also integrated for training the model. The output is the BC subtype that can further help clinicians confirm the diagnosis and decide the line of treatment. The framework for subtype classification is illustrated in Figure 1a. The performance of the proposed model is compared with classification based on single omics data and other state-of-the-art subtype classification methods.

### 1.2. Drug Response

One of the important aspects of personalized medicine is predicting drug response. Multiomics data have provided a better insight into cancer phenotypes [20] by highlighting the complexity of biomolecular data. Drug response is mainly influenced by genetic makeup. Predicting the sensitivity of a particular drug to BC cases is a crucial problem, as it is a baseline for the design of drugs. Clinicians are widely adopting it for treating cancer patients. The success of targeted therapy lies in identifying the target that may be a mutation or highly expressed gene. Based on the target, patients may be administered the right drug quantity to inhibit tumor growth. Identifying the drug and quantity is a problem that can be addressed by developing drug response prediction models for targeted therapy. Cancer cell-lines play an important role in such research, as insufficient data are available on patients with drug responses. Databases such as the Cancer Cell Line Encyclopedia(CCLE) and Genomics of Drug Sensitivity in Cancer (GDSC) provide omics data such as gene expression, mutation, and the methylation of cell lines and drug response matrix for cancer drugs. The drug response matrix indicates each drug versus the cancer cell line. The indicator may be a measure like Max IC50. These datasets can be integrated to investigate the drug response of various drugs on BC cell lines. Knowledge discovery of profiles linked to cancer and their interactions help in the identification of gene signatures with therapeutic impact [21].

Zhang et al. [22] built a similarity network to predict drug sensitivity. Their computational model was driven by gene expression of the cell lines and the molecular structure of drugs. Turki et al. [23] proposed a link prediction approach for predicting sensitivity to drugs. Their method first selects a subset of features using a matrix decomposition technique that is later trained using a supervised link prediction algorithm. Sharifi et al. [24] proposed a late integration multiomics drug prediction model based on deep neural networks. They validated their model for seven drugs that included two targeted and five chemotherapy drugs. The synergy of integrated omics-profiles along with drug features has not been exploited to the fullest for BC subtypes. Hence, we propose a late integration-based deep learning model: DCNN-DR, the Deep Convolution Neural Network-Drug Response model. The proposed model integrates omics data and leverages convolution neural networks to predict the response of up to 108 drugs on cancer cell lines using a threshold. We performed further analysis to predict the effective drugs for the BLBC subtype using the proposed model. The DCNN-DR model exhibits better performance than models based on single omics data. The model also fairs well compared to other state-of-the-art methods for certain drugs. The framework used for the problem is depicted in Figure 1b. These integrated omics-data-based predictions may help clinicians make reasonable decisions in therapeutics and provide insight into the biomarkers that drive cancers.

## 2. Materials and Methods

### 2.1. BC Subtype Classification

#### 2.1.1. Datasets

The multiomics data that is already preprocessed may be found on the FireBrowse portal [25]. The source hosts omics-datasets of more than a thousand BC patients’ information. The omics range includes GISTIC2 CNV, miRNA, mutation, RNA, protein expression and methylation data of the patients. The number of features of each dataset is shown in Table 2. Along with multiomics information, the clinical statistics of patients were also included in the study. The final dataset with 20% missing values filtered and imputed can be found in [26]. Two hundred and seventy-nine of the total patients had overlapping information of all the aforementioned six omics data along with clinical information. The number of samples of each subtype i.e., luminal A, luminal B, Basal-like, Her2+ and Unknown are 140, 68, 47, 15 and nine, respectively. The synthetic minority oversampling technique (SMOTE) [27] was used to handle the class imbalance problem. SMOTE is an oversampling technique that synthesizes samples for minority classes. In our case Luminal A and Luminal B samples outnumbered other subtype samples. This number was balanced by synthesized samples produced by SMOTE. The sampling process increased the number of observations from 279 to 600.

#### 2.1.2. Methods

In general, for a deep learning model, we need not choose features individually because the weights of the neural network can do it. Nonetheless, the “big p small n” paradigm in omics data [28], where p is the number of features and n is the number of samples, distort the deep learning model. To overcome this issue, we performed feature engineering before training the deep neural network [29]. We chose nearest component analysis (NCA) for feature reduction. NCA is a supervised feature selection technique that considers the effect of labels corresponding to the samples, making it a reasonable choice for the classifiers [30]. This is contrary to popular unsupervised methods like principal component analysis or t-Stochastic neighbor embedding which are blind towards observation labels. This feature engineering step helps in representing data in low dimension space without losing the data integrity in biological processes [31]. Given each omics profile of m instances, X = x1, x2, …, xm ∈ Rf and label for each instance is c1, c2 … cm, the data are represented by m x f matrix with n instances and f features. In our case each instance is patient information with f genes and each class label is BC subtype of the patient. The genes represent measurements of methylation, mRNA and other omics data. The core distance metric [32] used by NCA is as given below in Equation (1):d(x,y) = (x − y)T (z − y) = (Ax − Ay)T (Ax − Ay)(1)

The resulting feature number generated for each omics dataset was 75 for CNV data, 100 for mRNA data, 23 for methylation data, 4 for mutation data, 30 for miRNA data and 14 for protein data. The feature engineered data is available on [26]. Similar to the method discussed by Sharifi et al. [24], our deep neural network model consisted of seven encoding subnetworks, one for each dataset and a classification layer. The encoding subnets were employed using the encoding layer of the autoencoder framework. Each omics type has its feed-forward encoding subnetwork that is fully connected with rectified linear activation function. To regularize the model, dropout was used. Batch normalization enhanced the training process. After learning from the encoding subnetwork, the learned features were concatenated and formed the input to the classification layer. Suppose each omics set is of dimension m × fi (m instances and fi features each for ith omics dataset). Then, the resulting integrated input for the classification layer is of dimension m × F and represented by Equation (2) where ⊕ is concatenation operator.
m × F = m × f1 ⊕ m × f2 ⊕ m × f3 ⊕ m × f4 ⊕ m × f5 ⊕ m × f6 ⊕ m × f7(2)

The classification layer was used to predict subtypes of breast cancer using the concatenated input. This layer was used with dropout and weight decay for regularization. Softmax regression and cross-entropy were used for multiclass classification and loss, respectively.

#### 2.1.3. Performance Measurement

The accuracy of the model is evaluated using performance measures such as accuracy, F-measure etc. The model’s performance was compared with single omics data model and other state-of-the-art methods that used multi-omics data.

### 2.2. Drug Response Prediction

#### 2.2.1. Dataset

The GDSC website [32] hosts the information of both chemotherapy and targeted drugs. The IC50 values of cell line and drug pair <Ci, Dj> were chosen for our experiment. Ci here represents a single sample with features from integrated-omics profiles. Dj represents a single drug. The IC50 values of the drugs form the label for each <cell-line, drug> pair. The processed and imputed IC50 values of 42 BC cell lines and 100 drugs are available at [26]. The remaining eight drugs (from GDSC2) that were considered for the experiment had relevance to BC as suggested in the literature [33] and hence were included as part of the study. IC50 values are log-transformed. The omics data and corresponding features of 42 cell-lines are summarized in Table 3. For external validation, omics data of the BC patients available on TCGA were used. The four omics profiles, namely CNV, methylation, gene expression and mutation binary data used to train the model were used to predict drug response and IC50 values for TCGA-BC patients. The omics data of 607 BC patients with all the relevant omics profile and NCA features (used for training) and PAM50 subtype classification were filtered. The processed TCGA omics data for BC is available on the GDAC portal [25] and linked-omics [34].

#### 2.2.2. Methods

Feature selection was performed similarly to the steps mentioned in Section 3.1. The final feature numbers used for training the model were 15, 26, 26 and 108 for the mRNA, mutation, CNV and methylation datasets, respectively. Omics-specific subnetworks were tailored to handle the information of each omics profiles separately. Subsequently late-integration was implemented, where each subnetwork first learns a representation of itself and all the learned representations are later concatenated. The four concatenated subnetworks can be represented by Equation (3) where Sr, Sc, Sm, Sd are subnets to handle mRNA, CNV, mutation, and methylation data, respectively.
N = Sr ⊕ Sc ⊕ Sm ⊕ Sd(3)

The network N of combined features forms the basis of the CNN layer of the proposed regression model. The model predicts the IC50 value for each cell/patient-drug pair <Ci, Dj>. Repeated five-fold cross-validation was used to improve the estimated performance of the model. The optimized hyper parameters that helped tune the model were learned using hyperas, a Bayes-optimization variant supported my sklearn libraries in Python. Optimizers such as Adam, SGD, and RMSprop were tested, of which RMsprop gave a good prediction accuracy. The learning rates and batch normalization were tweaked to 0.002 and 128, respectively. Tanh activation was used on individual subnets and relu was used for the CNN layer. Dropout was used to prevent overfitting. The loss function used was a mean-squared error.

After predicting IC50 values for 108 drugs, the drugs with Pearson’s correlation coefficient (PCC) values less than 0.7 were discarded. Consequently, each cell line and drug pair was classified as sensitive or resistant based on preset thresholds. The median IC50 value for each drug was set as a threshold. The next important step is binary classification of the cell lines into two categories, sensitive or resistant, using the median IC50 for each drug as the threshold. Each cell line and drug pair <Ci, Dj> is marked as sensitive if the corresponding IC50 value of <Ci, Dj> is smaller than the threshold; otherwise, it is marked as resistant. Sensitivity suggests that the jth drug is effective and is a probable therapeutic candidate that can be considered for the treatment of the sample ith cell line, Ci.

#### 2.2.3. Performance Measurement

The performance of the multiple-output regression model was evaluated using metrics such as mean squared error and r2 score. PCC was calculated for real and predicted IC50 values for each <Ci, Dj>. The state-of-the-art methods were compared to check the effectiveness of popular cancer drugs on cell lines and BC patients. The performance in terms of the AUC of certain drugs was compared with the AUC published in the literature.

## 3. Results

### 3.1. BC Subtypes

The performance of the proposed classifier was evaluated for both binary and multiclass classification. Additionally, the performance was also compared with other similar approaches. Figure 2a shows the confusion matrix reflecting the performance of the classifier for each subtype.

The total number of test samples picked by the model were 140. Her2+ and unknown predictions were 100% accurate. Out of 34 basal-like subtype samples, one was misclassified as luminal B. The model incorrectly classified four instances of luminal A as luminal B. Additionally, three instances of luminal B were misclassified as luminal A. The confusion between luminal A and luminal B subtypes may be attributed to their molecular similarity; both ER and PR are highly expressed in these cases. Figure 2b (Appendix A) shows different metrics for comparing the performance of the proposed models with single omics data such as miRNA, methylation, mRNA and clinical data (Appendix A). The accuracy of the proposed multi-omics model with late integration was 0.94.

In Figure 2c (Appendix A), performance of the model for luminal A vs. all other subtypes is presented. The model’s performance deteriorates while classifying the luminal A and luminal B subtypes. Figure 2d compares multiomics models based on machine learning techniques such as random forest, elastic net and microkernel [13] learning. A state-of-the-art deep learning model DeepMO [18] was also compared with our model. Our integrative model performed fairly well compared to the existing methods.

#### Biological Relevance

The feature selection (FS) step can improve the accuracy of multiclass classification. The accuracies with FS and without FS are 0.79 and 0.94, respectively. The features from all of the omics datasets used in the training process were combined and ranked using the NCA technique. A list of the 20 top-ranked genes with their *p* values is presented in Appendix A. Metascape [35] analysis of the top seven screened genes, TP53, CDH1, EGFR, ANKS4B, B3GAT1, ESR1, and TMEM90A is shown in Figure 3a,b. Figure 3b is a network of the protein–protein interaction network reflecting pathways responsible for BRCA. Figure 3a,c,d present GO analysis and DisGeNET. These analyses show that genes that have an impact on the proposed model also have biological relevance.

### 3.2. Drug Response

Drug-response prediction is a multiple-output regression problem developed to predict IC50 values for a sample and drug pair. We trained the model using the omics data of 42 BC cell-line samples from CCLE. The NCA feature selection method aided in optimal feature selection for omics data, including information on 108 drugs from GDSC1 and GDSC2 and omics data for 42 BC cell-line samples. Both chemotherapy and targeted drugs were used for training the model. The resulting DCNN-DR model backed by hyperas optimization and five-fold cross-validation resulted in a regression value of 0.95 (Appendix A) and an average mean squared error of 0.63, which reflects the linear relation between the predicted and actual IC50 values. Predicted IC50 values for the drug, including omipalisib, gemcitabine, epothilone B, mitomycin C, luminespib, etc., exhibited a high correlation with real IC50 values, while the drug response of tamoxifen, niraparib, and JQ1 had the least correlation with their real counterparts. Appendix A list the performance measures of all drugs considered in the experiment. The overall classification accuracy after applying the median threshold is 0.80.

The proposed DCNN-DR model performed fairly well for many drugs compared to similar machine learning and deep learning (DL) models. Figure 4 indicates the values of different metrics used in evaluating the model’s performance. The values reflect that even with few samples and for a large group of compounds, the model’s performance did not degrade. Figure 4a,b captures the PCC and MSE of the regression model (Appendix A). Figure 4c shows the r-squared value of six cancer drugs. Of these six drugs, four are targeted drugs and are usually preferred over chemotherapy molecules mainly because they target only cancer cells. The classifier’s accuracy, sensitivity, and specificity are shown in Figure 4d (Appendix A). A comparison of the AUC of the drugs docetaxel and gemcitabine using the existing methods of Malik et al. [26] and MOLI [24] is shown in Figure 4e,f (with Appendix A). Additionally, the SVM-based model was implemented in-house, and its performance was compared with that of the proposed method. Figure 4g represents the sensitivity and resistance of cell lines to 15 drugs.

Appendix A shows the effectiveness of each drug over each cell line and the sensitivity of the cell line for each drug. Experiments show that the drugs with outliers had a negative impact on the model’s behavior (Appendix A). Alternatively, drugs with high correlations, such as Epothilone_B, Luminespib, XL_880, and PI_103, positively affected the model’s performance. The robustness of the DCNN-DR model is captured for a small subset of drugs with a high correlation. Almost 99 drugs showed a high correlation (Appendix A).

#### Clinical Significance

The model was successful in predicting effective targeted drugs for the Her2+ and luminal subtypes (Appendix A). Here, we will delve deeper into the BLBC subtype to check our model’s effectiveness. The BLBC subtype accounts for 15–20% of total BC cases [36]. This subtype exhibits high heterogeneity and the worst prognosis without any targeted therapy. The conventional treatment for BLBC is chemotherapy. Additionally, BLBC patients have poor survival because their tumours often have an incomplete pathological response to treatment. Studies have also shown that the EGFR gene is highly expressed in up to 78% of BLBC cases [37], but clinical trials have not shown great improvement using EGFR targeted therapy [36], mainly because EGFR downstream signalling pathways were still activated in most patients after EGFR-targeted treatment. This fact implies that there might be other pathways involved in bypass activation. As a result, EGFR-targeted treatment alone cannot achieve significant efficacy; instead, a combination of PI3K, MAPK, and Scr inhibitors may benefit growth factor inhibitors. The BLBC subtype has abnormal expression of MYC, PIK3CA, CDK6, AKT2, KRAS, FGFR1, IGF1R, CCNE1, CDKN2A/B, BRCA2, BRAF, PTEN, MDM2, RB1, TP53, EGFR, MET, NGF and HDAC1. Possible drugs for this subtype include growth factor inhibitors, DNA synthetic inhibitors, PARP inhibitors, genotoxic agents, mTOR inhibitors, histone deacetylase inhibitors, CDK inhibitors and other inhibitors depending on the target, as summarized in Table 4 [36,37,38]. The targeted drug selection delivery strategies require more clinical trial results to validate studies.

The DCNN-DR model was used to predict the effective drugs for 93 BLBC patients whose information is hosted in TCGA. The samples were organized as required by the model; i.e., four omics datasets, namely, mRNA, CNV, mutation and methylation data, and the same set of genes used to train the model were used for predicting IC50 values for each drug. Subsequently, the patients were categorized as sensitive or resistant to each drug. TCGA samples have PAM50 classification information of BC patients, as mentioned in Section 2. Ninety-three samples were of the BLBC type. The cell lines have 19 BLBC type samples that include basal-like A and B subtypes. Both subtype samples were combined into single basal-like samples for further analysis. Figure 5a,b show the top 35 and 26 effective drugs for cell lines and BC patients, respectively. Effective drugs are those that have 65% or more sensitive cell lines/patients. Figure 5a shows the real and predicted responses for the effective drugs. Effective drugs for TCGA samples are plotted in Figure 5b.

Table 5 shows the effective drugs predicted for cell lines and TCGA patients. Thirty-three of the total drugs listed are targeted drugs. Most of the abnormal genes identified for BLBC have targeted drugs. The predicted drugs are consistent with the literature for treating the BLBC subtype [38]. The proposed model may be clinically significant with more clinical trials.

## 4. Discussion

The classification of BC patients into proper subtypes is important in selecting the line of treatment. Moreover, having prior knowledge of how each subtype is distinguished is vital for research into new treatments. Although the existing IHC approach is useful, it is too broad [39]. Individuals with the same IHC subtype, for example, may not benefit from the same treatment regimens. Several studies have used omics data to find more specific subtypes of breast cancer. The use of omics data to create subtypes could lead to personalized treatment by identifying molecular profiles that are specific to certain subtypes. As a result, rather than relying only on IHC status, specialized treatments might be customized to these profiles in a more precise way.

In this work, we demonstrated that high classification accuracy can be achieved with the combination of feature selection and deep learning techniques. The model identifies Her2+, basal-like and unknown subtypes with great success, while luminal A and luminal B subtypes were misclassified in a few cases. Further analysis of the top-ranked genes identified by NCA showed their previously established role in cancers.

The drug response model developed for predicting effective drugs for cancer is vigorous enough to handle more than 100 drugs. DCNN-DR captured the association between the integrated omics profile and onco-drugs such as foretinib (r2 = 0.75), bleomycin (r2 = 0.77) and omipalisib (r2 = 0.87) (Figure 3c), among others, with a high degree of confidence. The R2 scores for some drugs were low due to outliers (Appendix A). Apart from predicting drug response for BC, the model can be extended conceptually for pancancer drug response prediction. These models may be incorporated into AI systems to complement the existing methods in health care. However, more clinical trials are needed for the models to be confidently accepted by clinicians.

We used our DCNN-DR model to predict effective drugs for BLBC cancer cell-lines and TCGA patients. The obvious reason for choosing this subtype for analysis is the complexity underlying this subtype. Since BLBC is a heterogeneous-target subtype, a combination treatment is suggested that includes a class of chemotherapy drugs along with targeted therapies based on the abnormal expression of specific genes or mutations. For the effective drugs predicted by the model for BLBC (Table 5), KEGG [40] and CMap [41] were used for further analysis. Target pathways and genes, especially small molecules, were identified using the above tools. The class of drugs identified for BLBC includes CDK inhibitors, histone deacetylase inhibitors, receptor tyrosine kinase inhibitors, BRAF inhibitors, mitosis inhibitors, DNA synthetic inhibitors, PARP inhibitors, growth factor inhibitors, mTOR inhibitors, etc. Recent investigations have also suggested the aforementioned inhibitors [37,42]. Whether a response to therapy is beneficial to the patient’s ultimate treatment routine is still a matter of debate. Exploratory analysis of BC subtypes and sensitivity analysis (Appendix A) to cancer drugs could open doors for practical solutions in improving therapies for cancer. Genomic profiling of cancer cell line panels and patient-derived samples has contributed greatly to building classification models and recommending novel remedies. Nevertheless, a pool of compounds has yet to be evaluated against available genomics data.

With highly available biological resources that capture disease characteristics such as phenotype, genotype and their relationships, novel approaches are indispensable. They will help in processing this information and discovering critical knowledge pertaining to the disease. We proposed late integrative deep learning frameworks for BC subtype classification and drug-response prediction models. Their performance is on par with existing individual solutions. We conclude that an artificial deep neural network, which is trained on the multiomics signature of an individual, in conjunction with its phenotypic factors, not only segregates BC patients to their subtypes but also assists in screening a pool of drugs based on the sensitivity values corresponding to the patient under observation. The results reinforce the idea that an integrative approach can make more accurate and personalized decisions for drug administration and general treatment strategies.

The proposed drug-response problem requires multiple outputs for each input sample, i.e., for each cell line/patient who the model predicts the response for more than one hundred drugs simultaneously. This is a multioutput regression problem that is inherently supported by deep learning neural networks such as CNNs. Although multioutput regression is built-in to machine learning methods such as the random forest, the links between inputs and outputs can be highly organized based on the training. This is a drawback of decision trees, especially for multioutput regression. Neural network models, on the other hand, have the advantage of learning from a continuous function that can model a more elegant relationship between input and output changes [43]. Another reason for choosing deep learning is to exploit the processing power of accelerators such as graphics processing units along with the readily available libraries to support their use. The proposed model can be extended for pancancer drug responses that require high processing power, making deep learning a good choice.

## 5. Conclusions

In this paper, we proposed two predictive models based on DL techniques and multiomics data to endorse personalized treatment: a model to predict the BC subtype of a patient and a model for drug response prediction, DCNN-DR. We trained the first model using BC patient omics and clinical data. The second model was trained using BC cell line omics data. We used the DCNN-DR model to predict the possible drugs for BLBC subtype patients. The performance of our models is similar to that of existing methods (Figure 2 and Figure 3), and they are clinically relevant. Nevertheless, more clinical trials are needed to attest the use of the proposed model. Approximately 5% of current treatments benefit from personalized treatment [24], and our work and other types of research are paving the way for more accurate therapeutics.

## Figures and Tables

**Figure 1 jpm-12-00674-f001:**
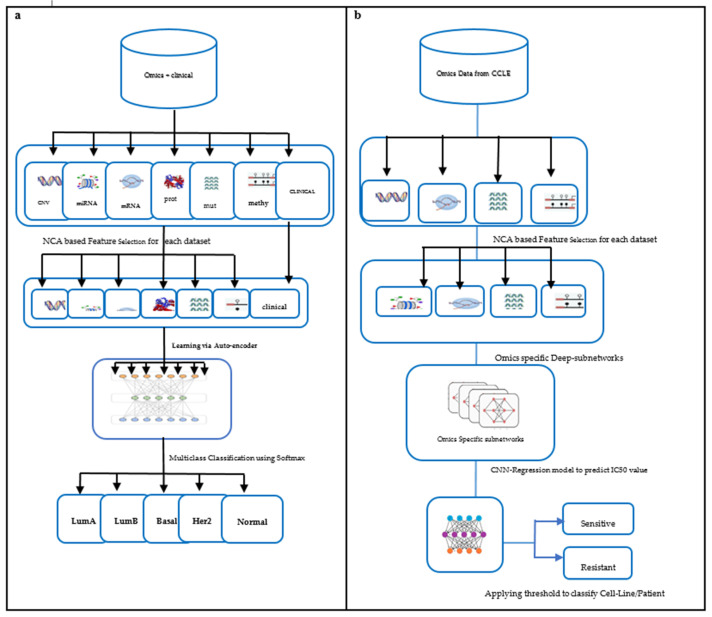
Framework for training predictive models: (**a**) Subtype classification. (**b**) Drug response prediction.

**Figure 2 jpm-12-00674-f002:**
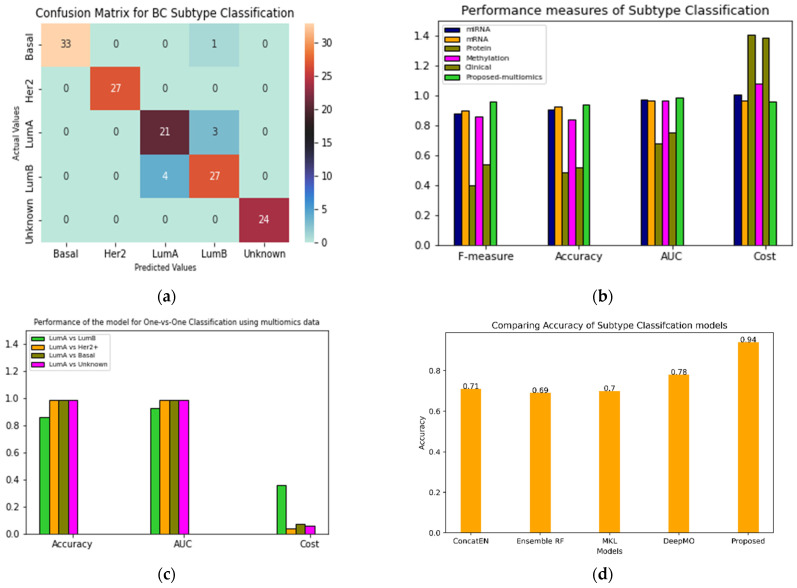
BC Subtype classification performance. (**a**) Confusion matrix for 140 test samples. (**b**) Single vs. multi-omics data performance comparison (**c**) Binary classification with Luminal A as common subtype in each case. (**d**) Comparison of existing methods; concatenated elastic net and random forest are in-house methods. DeepMO and MKL are mentioned in the literature.

**Figure 3 jpm-12-00674-f003:**
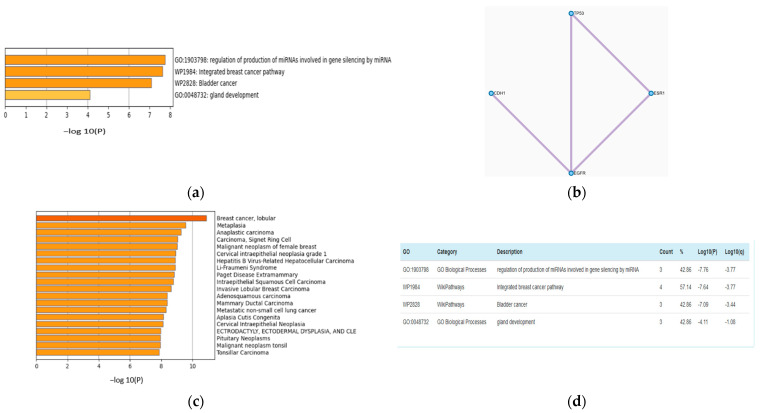
Analysis of genes from mutation omics data using Metascape. (**a**) GO Analysis. (**b**) Protein–protein interactions of the shortlisted genes. (**c**) DisGeNET analysis reflecting the top ranked gene contributions to BC. (**d**) Wiki and Go pathway results.

**Figure 4 jpm-12-00674-f004:**
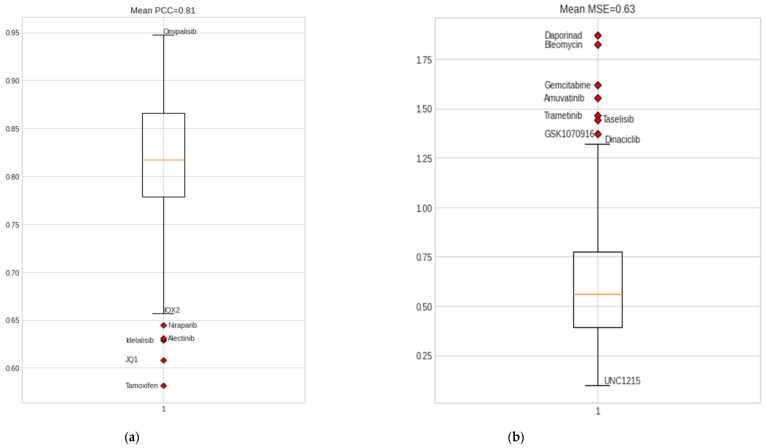
Performance of DCNN-DR using various metrics. (**a**) PCC for individual drugs. (**b**) Mean squared error of each drug. (**c**) Performance evaluation of the model using r2_squared error as a metric for six cancer drugs. (**d**) Box plot showing the accuracy, sensitivity and specificity of classification for all drugs. (**e**) AUC of different techniques for docetaxel and (**f**) gemcitabine. (**g**) Violin graph representing the sensitivity and resistance of cell-line samples for fifteen cancer drugs.

**Figure 5 jpm-12-00674-f005:**
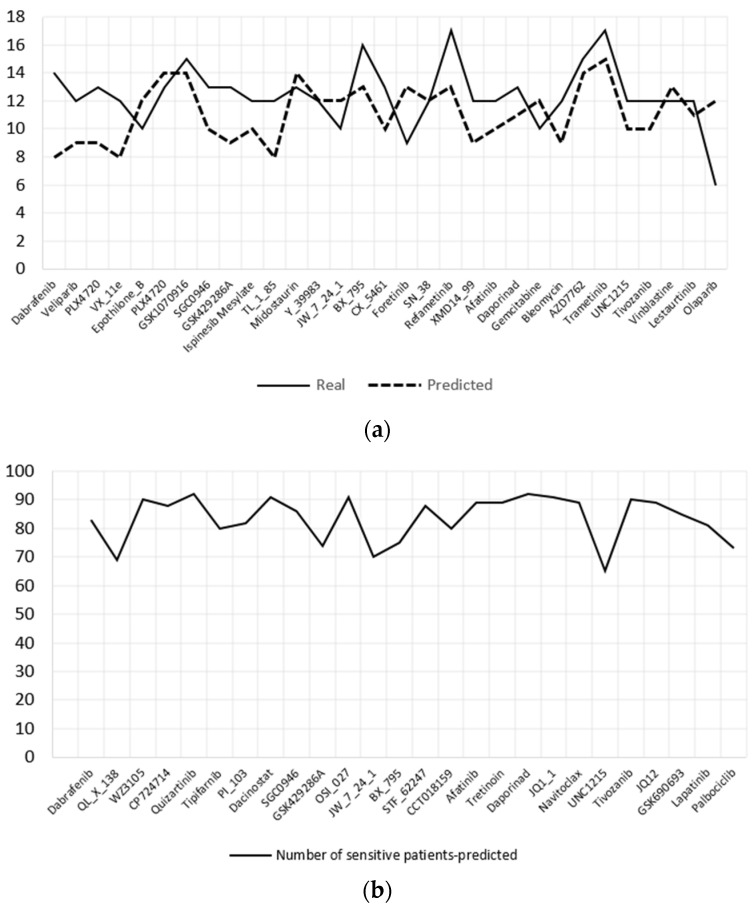
Effective drugs identified by the model for BLBC subtype. (**a**) Top 31 drugs for BLBC cell lines both predicted and real sensitive cell lines are shown. (**b**) Predicted 26 effective drugs for BLBC patients.

**Table 1 jpm-12-00674-t001:** PAM50 BC subtype classification.

BC Subtypes	Suggested Therapies and Other Information
Luminal A	Endocrine Targeted Therapy; Low grade cancers with better survival rate
Luminal B	Targeted Endocrine Therapy; Elderly patients are affected with prognosis slightly worse than Luminal A subtype
Basal-like	Chemotherapy. No targeted therapy; Poor prognosis Is further classified as Basal-like type 1 and 2 etc.,
Her2-enriched	Her2 Targeted therapy; Poor prognosis
Normal-like	Targeted therapy that targets Ki-67; Good prognosis

**Table 2 jpm-12-00674-t002:** Summarized information of omics data used to train the model.

Omics/Phenotype Data	Total Number of Observations	Number of Features Originally	Description
RNA	1093	17,814	z-scaled RSEM values
miRNA	1078	1046	log2-RPM value
Mutation	977	977	Binary data for gene mutation
CNV	1089	15,186	Values computed from patient’s GISTIC2
Methylation	1097	27,578	DNA methylation Scaled β values
Protein	887	226	Scaled β values
Clinical Data	1097	19	Only clinical features like age, days to the last follow-up, gender, lymph node metastasis, the number of affected lymph nodes, pathologic stage, tumour stage, histological type and metastatic stage were considered.

**Table 3 jpm-12-00674-t003:** Number of features for each omics dataset for 42 cell-line samples.

Omics Data	Total Number of Features for Each Omics
mRNA	697
Mutation	34,673
CNV	710
Methylation	808

**Table 4 jpm-12-00674-t004:** Summary of therapeutic strategies and target pathways for BLBC.

Therapeutic Target Strategies	Target Pathways
Inhibit cell proliferation	Mitosis
Inhibit DNA damage response	DNA replication
PARP inhibitors	PI3K/mTOR signalling
EGFR inhibitors	Growth Factor inhibitors
MET inhibitors	mTOR signalling
CDK inhibitors	PI3K-Akt signalling pathway
BRAF, MEK1, MEK2 Inhibitors	ERK/MAPK signalling
Histone deacetylase inhibitor	Notch signalling pathway
Receptor tyrosine kinase inhibitor	VEGF/IGF-1R pathways

**Table 5 jpm-12-00674-t005:** Summary of effective drugs predicted by DCNN-DR.

Sl No.	Top Effective DRUGS as Predicted by Proposed Model	Targets	Target Pathway
1	Bleomycin(ct)	dsDNA break induction, DNA	DNA replication
2	Gemcitabine	Pyrimidine antimetabolite	DNA replication
3	Mitomycin-C	DNA crosslinker	DNA replication
4	SN-38	TOP1	DNA replication
5	Afatinib	ERBB2, ERBB4, EGFR	EGFR signaling MAPK signaling pathway, ErbB signaling pathway,
6	Dabrafenib	BRAF	ERK MAPK signaling, MAPK signaling pathway ErbB signaling pathway
7	HG6-64-1	BRAF, ERBB4, FGR, MAP3K9, AURKC	ERK MAPK signaling
8	PLX-4720	BRAF	ERK MAPK signaling
9	Refametinib	MEK1, MEK2	ERK MAPK signaling
10	Trametinib	MEK1, MEK2	ERK MAPK signaling, ErbB signaling pathway
11	Omipalisib	PI3K (class 1), MTORC1, MTORC2	PI3K/MTOR signaling
12	OSI-027	MTORC1, MTORC2	PI3K/MTOR signaling
13	Daporinad	NAMPT	Metabolism
14	Docetaxel	Microtubule stabiliser	Mitosis
15	Epothilone B	Microtubule stabiliser	Mitosis
16	GSK1070916	AURKA, AURKC	Mitosis
17	Ispinesib Mesylate	KSP	Mitosis
18	Vinblastine	Microtubule destabiliser	Mitosis
19	Olaparib	PARP1, PARP2, BRCA	Base excision repair, NF-kappa B signaling pathway
20	Navitoclax	BCL2, BCL-XL, BCL-W	Apoptosis regulation
21	AZD7762	CHEK1, CHEK2	Cell cycle
22	Belinostat	HDAC1	Chromatin histone acetylation, Cell cycle, Notch signaling pathway,
23	Dacinostat	HDAC1	Chromatin histone acetylation
24	JW-7-24-1	LCK	MAPK signaling pathway
25	CX-5461	RNA Polymerase 1	ATM/ATR pathway
26	Midostaurin	PKC, PPK, FLT1, c-FGR, others	MAPK signaling pathway, PI3K-Akt signaling pathway, VEGF signaling pathway, Leukocyte transendothelial migration
27	Tipifarnib	Farnesyl-transferase (FNTA)	Terpenoid backbone biosynthesis
28	WZ3105	SRC, ROCK2, NTRK2, FLT3, IRAK1, others	NF-kappa B signaling pathway
29	BX795	TBK1, PDK1 (PDPK1), IKK, AURKB, AURKC	NOD-like receptor signaling pathway
30	Lestaurtinib	FLT3, JAK2, NTRK1, NTRK2, NTRK3	MAPK signaling pathway, PI3K-Akt signaling pathway
31	QL-X-138	BTK	Tyrosine kinase pathway
32	Ruxolitinib	JAK1, JAK2, JAK3, TYK2	Chemokine signaling pathway, JAK-STAT signaling pathway,
33	Luminespib	HSP90	PI3K-Akt signaling pathway, Estrogen signaling pathway, Protein stability and degradation
34	Foretinib	MET, KDR, TIE2, VEGFR3/FLT4, RON, PDGFR, FGFR1, EGFR	RTK signaling, VEGF signaling pathway, Focal adhesion
35	Lapatinib	CYP3A5, EGFR, ERBB2	MAPK signaling pathway, ErbB signaling pathway, Breast cancer
36	Tivozanib	VEGFR1, VEGFR2, VEGFR3, FLT1, FLT4, KDR, KIT, PDGFRA, PDGFRB	MAPK signaling pathway, Ras signaling pathway, Rap1 signaling pathway, PI3K-Akt signaling pathway, VEGF signaling pathway, Focal adhesion
37	PD173074	FGFR1, FGFR2, FGFR3	MAPK pathway
38	NU7441	DNAPK	DNA repair pathway

## Data Availability

All the codes and data used for the study can be accessed at https://github.com/Deebamajeed/BC-Subtype-DR (accessed on 6 March 2022).

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
