# Peer review of "Leveraging Deep Learning Techniques and Integrated Omics Data for Tailored Treatment of Breast Cancer"

_jpm, 2022, doi:10.3390/jpm12050674_

Round 1
Reviewer 1 Report
Comments:
- Some description sentences need to be rephrased, such as Line 119 - 122. Please avoid directly using the reference index as part of a sentence. Please identify the research group or methodologies explicitly.
- Please provide a full name, brief introduction, and references while mentioning an existing work or method. The description such as "SMOTE was applied to handle class imbalances" in Line 147 is difficult for the audience to follow and reproduce the results if needed.
- The diagram showing the architecture and prediction workflow is very informative. It will be better if the authors could provide more discussion and demonstration to justify such designs. The questions to address include:
1) What is the purpose of feature selection? Deep learning models are usually considered robust end-to-end methods. With proper regularization, the feature selection can be accomplished by the neural network itself. Additional feature selection steps are usually not conducted.
2) Autoencoder is a typical unsupervised learning method. According to the diagram, it works with a downstream multiclass classifier, which is apparently a supervised learning model. What is the purpose of using autoencoder structure in this architecture? It will also be helpful if the authors could clarify which part of the autoencoder is used as the input of the classifier. If the latent space was used, were there any extra processing steps applied, such as normalization?
3) For dimension reduction and feature selection purposes, could the authors justify the usage of NCA? Why does NCA outperform other methods such as PCA or t-SNE? - For the drug response regression task, could you describe the construction process of omics subnetworks? Why 1-D CNN was used as the regression model? What is the most unique characteristic of this model so that it outperforms the other simpler regression models?
Author Response
File attached

Reviewer 2 Report
In the submitted manuscript “Leveraging deep learning techniques and integrated omics data for tailored treatment of BRCA” the authors presented an interesting approach to determine a tailored treatment for the selected group of cancer patients.
But the study is flawed with several minor and major issues. I have highlighted some of the issues here and hope it would be beneficial in improving the predictive model.
Line 246 – Typographical error. Mutation in “GAPDH”. Also, GAPDH is considered widely as a house-keeping gene and one of the most stable in both RNA and protein level. It is doubtful that there will be too many mutations associated with this gene. Furthermore, the training set of genes chosen from the Metascape analysis, viz. TP53, PIK3CA, GAPDH, NFKB1, PGR and HSPA1 are so widely used in various cellular pathways, it is not surprising that these gene cluster is recognized to be associated with various cancer pathways by the GO and KEGG analysis. The real biological significance of the process is still missing.
Regarding Drug response predictions, the authors did not highlight why the model predicted IC50 for many drugs did not correlate with the real IC50. The drawback of the model is needed to be critically evaluated to improve the model.
Regarding the clinical significance of the model, the authors highlighted the possibilities of using drugs such as “growth factor inhibitors, DNA synthetic inhibitors, PARP inhibitors, genotoxic agents, mTOR inhibitors etc”. These are known drugs for cancers for ages; but the problem with these drugs is that they kill cancer cells as much as the normal cells. The model is no longer helping in selection of the choice of drugs in this case.
Overall, the model needs to be rigorously tested with the clinical outcomes and trained with stringent parameters. Otherwise, the model predicted tailored treatments are not tailored enough for any particular kind of cancer treatment.
Author Response
File attached

Reviewer 3 Report
The authors present two deep learning- based models that may be useful in the treatment of breast cancer patients. The first model can be used for breast cancer molecular subtype classification whereas the second model predicts the effectiveness of therapeutic drugs in vitro and in vivo. They also test the predictive performance of this model in distinguishing the most effective drugs for basal like breast cancer.
Deep learning based models are evolving rapidly and will play a key role in the fight against cancer. The concept of the study is very interesting and the models are characterized in detail. Unfortunately, the manuscript is poorly written and quite confusing and it was impossible for me to follow the presentation of the results and the discussion of findings. Even the use of term BRCA for breast cancer is confusing, because BRCA is usually used for BRCA-mutated inherited breast cancer.Instead, I would suggest the abbreviation BC.
Question:
The classification of breast cancer molecular subtypes is currently based on immunohistochemical evaluation of ER,PR,HER2 and Ki67 in biopsy/resection specimen. Immunohistochemistry is a cheap and robust technique, therefore what is the clinical significance of the predictive model for subtype classification?
Author Response
File attached

Reviewer 4 Report
The authors proposed two predictive models to endorse the personalized treatment: a model to predict the BRCA subtype of a patient and a model for drug response prediction, DCNN-DR. They trained the first model using BRCA patients’ omics and clinical data. The second model was trained using BRCA cell-lines omics data. The author demonstrated that approximately 5% of current treatments are benefitted from personalized treatment. Albeit, the current study paves the way for more accurate therapeutics. I still have some minor suggestions.
1, All figures are highly professional, and the authors should guide the readers to the meaning of the images appropriately; otherwise, it is likely to cause misunderstandings. Therefore, I suggest that the author consider revising these figure legends again.
2, In Table 5, the author summary of effective drugs predicted by DCNN-DR. Since Connectivity Map (CMap) can be used to discover the mechanism of action of small molecules, functionally annotate genetic variants of disease genes, and inform clinical trials (PMID: 29195078). Is this possible that the author can also perform data mining and validate their data via CMap , and make some comments in the Discussion?
3, There are a few typo issues for the authors to pay attention. Please unify the writing of scientific terms. “Italic, capital” ? make it consistent throughout the whole manuscript. For example, line 13, in-vitro and in-vivo should be “Italic”. Line 26,… data suggests that,” Female “….
4, Line 69-70, TNBC: ER+/PR+/HER2-, HER2+ and TNBC. Please correct this sentence.
5, The author needs to follow the “Instructions for Authors” and edit the format of the references for the manuscript (https://www.mdpi.com/journal/jpm/instructions#references).
6, Please try to avoid using unimaginably long sentences, such as: Line 119-123, and the figure manuscript also need English proofreading.
Author Response
File attached

Round 2
Reviewer 3 Report
Although the authors did improve the manuscript according to the suggestions of the reviewers, i believe that it cannot be published in the current form.
The breast cancer subtype classification part is still not convincing, mainly due to the unclear presentation of data. For example, how many patients are included in each subtype group? What is the performance of the model in differentiation between two subtypes i.e luminal A versus luminal B, luminal A versus TNBC, luminal A versus HER2 (+) etc. This information is not present in the manuscript.
The proposed model is interesting and may have clinical application in the future but this should be demonstrated by the authors in a more clear and convinving way.
Author Response
Doc attached
